# Investigating the use of a one-page infographic to improve recruitment and retention to the BASIL+ randomised controlled trial: A Study Within a Trial (SWAT)

Lucy Atha[1]*, Eloise Ryde[1,2], Lauren Burke[1], Samantha Brady[1], Kalpita Baird[1]
Leanne Shearsmith[3], Charlie Peck[1], Andrew Henry[1,2], Caroline Fairhurst[1], Han-I Wang[1],
Della Bailey[1], Rebecca Woodhouse[1], Dean McMillan[1,4], Simon Gilbody[1,4], David Ekers[1,2],
Elizabeth Littlewood[1,2]

1 Department of Health Sciences, University of York, York, United Kingdom, 2 Tees, Esk and Wear Valleys NHS FT, Research & Development, Flatts Lane Centre, Middlesbrough, United Kingdom, 3 Leeds Institute of Health Sciences, University of Leeds, Leeds, United Kingdom, 4 Hull York Medical School, University of York, York, United Kingdom

* lucy.atha@york.ac.uk

## Abstract

### Background

Low participant recruitment and retention rates are a significant barrier to successful Randomised Controlled Trials (RCTs). A Study Within A Trial (SWAT) is an effective way to explore which trial delivery methods may be useful for improving participant recruitment and retention rates. Infographics are a useful information delivery tool that may improve participants' understanding of the trial and thus improve recruitment or retention rates. This SWAT was embedded within the Behavioural Activation in Social Isolation (BASIL+) RCT. BASIL+ was delivered during the Covid-19 pandemic and evaluated the clinical and cost-effectiveness of a brief psychological intervention (Behavioural Activation) to mitigate depression and loneliness in older adults with multiple long-term health conditions.

### Methods

Twelve research sites were randomly allocated 1:1 to either the SWAT intervention group (participant information included a one-page infographic) or the SWAT control group (participant information did not include a one-page infographic). The primary outcome was the recruitment rate to the BASIL+ trial. The secondary outcomes were the number of expressions of interest in the trial and the follow-up retention rate at 3 months post-randomisation. Results were compared for each group using a mixed-effect logistic regression model with trial site as a random effect. The cost-effectiveness of the SWAT intervention was also evaluated.

**Data availability statement:** All relevant data are within the manuscript.

**Funding:** DM, SG, DE and EL were funded by the National Institute for Health and Care Research (NIHR) Programme Grants for Applied Research (PGfAR) RP-PG-0217-20006. The funder had no role in study design, data collection and analysis, decision to publish, or preparation of the manuscript.

**Competing interests:** The authors have declared that no competing interests exist.

## Results

Despite a small additional cost (£0.13) per participant, there was no evidence that participant recruitment, expressions of interest or retention was significantly affected by the inclusion of the one-page infographic.

## Conclusion

Our results suggest that the inclusion of an infographic alongside the participant information sheet may not be the best way to improve recruitment and retention rates for RCTs. However, infographics continue to be effective tools for information delivery in healthcare settings, and further research is needed to explore their use in RCTs.

---

## Introduction

It is well known that randomised controlled trials (RCTs) often experience difficulties in recruiting the target number of participants. For example, a review of 151 RCTs funded by the UK National Institute for Health and Care Research (NIHR) Health Technology Assessment (HTA) programme revealed that only 56% of the studies reached their recruitment targets [1]. This is a critical problem in research, as without an adequate number of participants, an RCT will be unable to provide meaningful results. Participant retention rates can also pose a significant barrier for RCT success, as inadequate retention rates, particularly at the primary outcome timepoint, can lead to statistically underpowered results. For example, missing just 20% of participant outcome data has been shown to significantly reduce the validity of an RCT [2]. To overcome these issues, strong recruitment and retention strategies are essential for ensuring robust results.

Using a Study Within a Trial (SWAT) is a valuable way to evaluate RCT procedures [3]. SWATs provide evidence of whether strategies aimed at improving recruitment and retention rates in RCTs are effective or not. Although individual SWATs lack statistical power, combining the evidence from multiple SWATs can provide meaningful results and reveal which strategies may be most effective, informing future RCT design [3].

When addressing the challenge of RCT recruitment and retention, participant information material is clearly an important area of study. For example, previous research has identified the participant information sheet (PIS) as a key area of focus for improving recruitment rates [4] and retention rates [5].

Previous research has explored various aspects of participant information materials for improving recruitment and retention rates. For example, studies suggest that changes to the length of the PIS or incorporating participant feedback when designing the content may have little or no effect on trial recruitment or retention [6–8]. However, research focusing on the layout and style of the PIS suggests that using colour and photographs and including a 'study timeline' flowchart may significantly increase the number of patients who attend a screening visit [9]. Researchers also found that during focus groups with non-participants, the majority of people preferred more

colourful and image-based participant information documents [9]. This suggests that incorporating more graphics-based information holds promise for improving both recruitment and retention rates in RCTs.

Infographics (information graphics) present information visually using graphics with minimal text. There is persuasive evidence that infographics are useful when delivering information to patients in a healthcare setting, improving comprehension and recall across age groups [10,11]. Research has also found that although infographics were no better than plain language text for increasing knowledge, they did make information more easily understandable and user friendly [12]. This evidence suggests that using an infographic could provide a valuable tool for communicating study information to potential trial participants in a clear and concise way, thus improving participant understanding of and interest in the trial and, in turn, recruitment and retention rates.

Despite the growing interest in the use of infographics in trials, no studies to date have thoroughly evaluated their cost-effectiveness. Given the potential resource implications of developing and implementing infographics at scale, it is important to assess whether any improvements in participant recruitment or retention justify the additional costs.

In this SWAT, we therefore aimed to evaluate the effectiveness and cost-effectiveness of using a one-page infographic alongside the standard text-based PIS for improving recruitment and retention to the BASIL+ RCT [13,14]. Our research question was "Does providing a one-page infographic alongside a written participant information sheet increase recruitment and retention rates compared to providing a written participant information sheet only?"

## Methods

### Trial design

We conducted a two-arm cluster-randomised SWAT, embedded within the BASIL+ RCT (ISCRN63034289) [13,14]. Twelve BASIL+ trial sites were randomised 1:1 to either the SWAT intervention group or the SWAT control group (one-page infographic plus PIS vs PIS only). The SWAT was approved by Yorkshire and the Humber–Leeds West Research Ethics Committee on December 11, 2021 (Ref: 20/YH/0347).

The BASIL+ RCT aimed to evaluate the clinical and cost-effectiveness of a remotely delivered psychological intervention to mitigate depression and loneliness in older adults with multiple long-term health conditions during isolation. The intervention was a form of support based on Behavioural Activation (BA) [15], set within a collaborative care framework, and was adapted for remote delivery during the Covid-19 pandemic [16]. BA support was provided to intervention participants over (up to) 8 weekly telephone sessions, aided by a self-help booklet. The comparator was usual care, where any care usually received by the participant continued, alongside providing signposting information with advice on keeping mentally and physically well. The BASIL+ outcome measures were collected at baseline, 1, 3 and 12-months post-randomisation in the form of questionnaires. The primary outcome was self-reported depression severity measured by the Patient Health Questionnaire (PHQ9) [17] at 3 months post-randomisation. Secondary outcomes included depression at 1 and 12 months, and loneliness [18] social isolation [19]; anxiety [20], and health-related quality of life [21] at 1, 3 and 12 months. Health service use data was also collected at all time points to evaluate the cost-effectiveness of the BA intervention. Further details can be found in the BASIL+ protocol [13].

### Participants

Patients registered with one of 26 participating NHS General Practices in England and Wales (each affiliated with one of the 12 BASIL+ research sites) were sent information about the BASIL+ trial between 08/02/2021 and 28/02/2022. To receive the study information, patients had to be aged 65 years or over and have two or more long-term health conditions, or a condition indicating they may be 'clinically extremely vulnerable' in relation to Covid-19 [22]. To be eligible to participate in the BASIL+ trial, patients also needed to score 5 or more on the PHQ9, indicating that they had depressive symptoms or were at risk of developing clinical depression. All patients receiving the study information via their general practice across all 12 BASIL+ research sites were therefore eligible for inclusion in the BASIL+ SWAT.

## SWAT interventions

Working with the BASIL Patient and Public Involvement Advisory Group (PPI AG), the BASIL+ study team created a bespoke one-page infographic (Fig 1), providing a visual representation of what participating in the BASIL+ study would involve. The PPI AG was closely involved in the development of the infographic from concept to completion. Due to the social distancing restrictions in place at the time of designing the infographic, PPI AG meetings were conducted via video call. Initial meetings involved discussing the infographic idea and determining it's format and content. Subsequent meetings focussed on refining drafts of the infographic, discussing the level of detail needed, as well as providing feedback on the colours and graphics used in the final document. For example, the addition of the arrows on the infographic was suggested by the PPI AG, as well as reducing the amount of text included. PPI AG members were reimbursed for their time according to INVOLVE guidelines [23]. Details of PPI AG involvement in BASIL+ are provided in the main BASIL+ findings paper [14].

All eligible SWAT participants were sent a study information pack containing a general practice letter-headed invite letter, the BASIL+ PIS and an example consent form. Those patients registered with general practices affiliated to research sites allocated to the SWAT intervention group also received the one-page infographic as part of the study information pack. The BASIL+ PIS explained the study information as written text, with no images.

## Outcomes

The primary outcome of this embedded SWAT was the participant recruitment rate into the BASIL+ trial. The recruitment rate is defined by the proportion of participants randomised to the trial relative to the number of SWAT participants who were sent the study information.

The secondary outcomes were the response rate to the information pack sent out, the retention rate of participants in the BASIL+ trial and the cost-effectiveness of the SWAT intervention. The response rate was measured by the number of expressions of interest received from potential participants in proportion to the number of participants who were sent the study information. Expressions of interest could be in the form of submission of an online consent form, direct contact via email or telephone with the local study team, or verbal permission given by the participant to be contacted by the study team. The retention rate was measured by the number of completed three-month post-randomisation participant questionnaires in proportion to the number of three-month questionnaires due.

SWAT intervention cost effectiveness was measured by the incremental cost per additional participant recruited, as well as the extra cost of the SWAT intervention overall and per-participant.

## Sample size

The sample size for the SWAT depended on the number of participating research sites open to recruitment, the number of general practices engaging in participant recruitment activities via each research site, and the number of eligible SWAT participants who were sent the study information via their general practice. Therefore, no formal sample size calculation was undertaken.

## Randomisation

BASIL+ research sites were randomised to a SWAT group on a 1:1 basis using simple randomisation in Stata v16 (StataCorp. 2019. Stata Statistical Software: Release 16. College Station, TX: StataCorp LLC.) before recruitment began. The BASIL+ trial statistician randomised the BASIL+ sites (using pseudonymised trial site ID number rather than site name) in one batch in December 2020 and passed the allocations to the trial team to distribute study information packs according to SWAT allocation group. Knowledge of SWAT group allocation was limited to key members of the BASIL+ trial team who were not directly involved in BASIL+ participant recruitment or completion of trial follow ups. Researchers were

# WHAT WOULD TAKING PART IN THE BASIL⁺ STUDY INVOLVE?

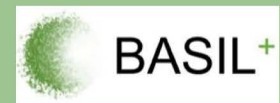

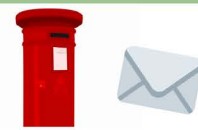

**1  RECEIPT OF STUDY INFORMATION**

- You receive study information in the post via your GP.

**CONTACT ABOUT THE STUDY**

**2**

- Someone from the practice team will call you to briefly explain the study. You can ask them questions.
- If you are interested, they will arrange for a BASIL⁺ researcher to contact you to ask you some questions.
- If you are not interested, you will not be contacted again.

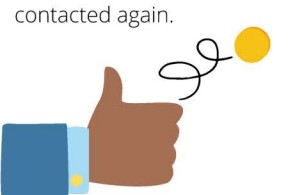

**3  GROUP ALLOCATION**

- If you are interested and able to take part, you will be enrolled into the study and a BASIL⁺ researcher will complete a questionnaire with you over the telephone.
- A computer system will then place you in one of two groups at random; this is like flipping a coin to decide.

**WHAT ARE THE TWO STUDY GROUPS?**

**4**

- **Usual care with signposting group** - you will continue with your care as usual and you will be signposted to information about maintaining your health and wellbeing.

- **BA support group** - you will receive a BASIL⁺ booklet and will be contacted by a BASIL⁺ Support Worker to arrange your first (of up to eight) BA support sessions.

What is Behavioural Activation (BA)?

BA is a form of support which aims to help people maintain or introduce activities which are important to them, and which may benefit physical and emotional wellbeing.

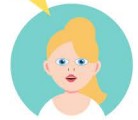

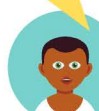

**5  FOLLOW UP CALLS AND FEEDBACK**

- You will be asked to complete a questionnaire 1 month, 3 months and 12 months after you join the study.
- You can complete the questionnaire over the telephone with a BASIL⁺ researcher or online.
- If you would like to, you can also agree to provide some feedback about being involved in the study.

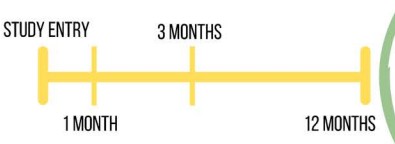

STUDY ENTRY    3 MONTHS

1 MONTH    12 MONTHS

**END OF PARTICIPATION**

- You will receive a summary of the study findings when these are available.

**6**

BASIL MT SWAT Infographic V1.0 4th January 2021

**Fig 1.  BASIL+ SWAT infographic.**

not aware of which SWAT group their research site was allocated to and had no access to the study information packs before or after they were sent to participants.

## Statistical methods

The BASIL+ SWAT data was collected at the York Trials Unit, in the form of randomisation and retention data from trial databases. Analyses were conducted in Stata v17 (StataCorp. 2021. Stata Statistical Software: Release 17. College Station, TX: StataCorp LLC.).

The primary outcome data was analysed using a mixed-effect logistic regression model with trial site as a random effect. The treatment effect is expressed as an adjusted odds ratio (OR) with 95% confidence interval (CI) and p-value.

The secondary outcomes were analysed in a similar way to the primary outcome, with the retention rate outcome also using main trial allocation (not recruited/control/intervention) as a covariate.

## Economic analysis

An economic analysis was conducted to evaluate the cost-effectiveness of the SWAT intervention. Cost data were collected using a bottom-up costing method, which included team member salaries associated with the infographic design, as well as additional costs for printing and postage. All costs were expressed in 2022 GBP values. Incremental effectiveness was measured using the absolute risk reduction (ARR), calculated as the difference between the percentage of participants who were recruited/randomised with the infographic (X%) and those recruited/randomised without it (Y%). Cost-effectiveness was then reported as the incremental cost-effectiveness ratio (ICER), defined as the incremental cost per additional participant recruited (incremental cost divided by incremental effectiveness). The analysis also reported the overall extra cost of the SWAT intervention as well as the cost per-participant.

## Results

Twelve research sites were randomised into the SWAT in December 2020; six to each arm. Recruitment to the BASIL+ RCT took place between 08/02/2021 and 28/02/2022, with a total of 11,944 study information packs sent out (6,136 with the infographic and 5,808 without the infographic). Expressions of interest were received from 1,325 potential BASIL+ participants across the 12 sites (11.1%; infographic, n = 626/6,136, 10.2%; no infographic, n = 699/5,808, 12.0%) and a total of 435 participants were randomised (3.6%; infographic, n = 189/6,136, 3.1%; no infographic, n = 246/5,808, 4.2%) (Table 1). The adjusted OR for the primary outcome was 0.62 (95% CI 0.23 to 1.66, p = 0.34) and for expression of interest was 0.61 (0.14 to 2.67, p = 0.52). A total of 359 randomised participants were retained in the trial at 3 months (3.0% of all potential participants approached; infographic, n = 165/6,136, 2.7%; no infographic, n = 194/5,808, 3.3%). While retention was higher in the no infographic group, the adjusted OR actually favours the infographic group as it adjusts for an imbalance in retention rates observed between the main trial allocation groups.

Despite sending out 885 study information packs, one research site received no expressions of interest or recruited any participants. Post-hoc sensitivity analyses were conducted excluding this site (Table 2) and the findings mirrored the main SWAT analysis.

**Table 1. Summary of the SWAT trial results.**

| Outcome, n (%) | Infographic (n = 6,136) | No Infographic (n = 5,808) | Odds Ratio (95% CI) | p-value |
|---|---|---|---|---|
| Recruited | 189 (3.1) | 246 (4.2) | 0.62 (0.23, 1.66) | 0.34 |
| Expression of Interest | 626 (10.2) | 699 (12.0) | 0.61 (0.14, 2.670) | 0.52 |
| Retention at 3 Months | 165 (2.7) | 194 (3.3) | 2.02 (0.93, 4.41) | 0.08 |

**Table 2. Summary of SWAT trial results excluding site which received no expressions of interest.**

| Outcome, n (%) | Infographic (n = 5,251) | No Infographic (n = 5,808) | Odds Ratio (95% CI) | p-value |
|---|---|---|---|---|
| Recruited | 189 (3.6) | 246 (4.2) | 0.91 (0.53, 1.54) | 0.71 |
| Expression of Interest | 626 (11.9) | 699 (12.0) | 1.21 (0.58, 2.49) | 0.61 |
| Retention at 3 Months | 165 (3.1) | 194 (3.3) | 2.02 (0.93, 4.41) | 0.08 |

No harms or unintended effects were reported.

## Economic analysis

The total costs for designing, printing and posting the infographic are outlined in Table 3. As shown, the total incremental cost of delivering the SWAT intervention was £821.52, resulting in an additional cost of £0.13 per participant in the infographic group, compared to control group participants who received no infographic. Furthermore, the average cost per recruited infographic group participant was £3.61 (£0.13 × 100/3.6) versus £0 in the control group (£0 × 100/4.2). The intervention was associated with higher incremental costs (£821.52) and a lower incremental recruitment rate (0.6% [3.6% vs 4.2%]) than the control. This indicates that the infographic intervention may not be a preferred option, as it is likely to be more costly without clear evidence of improved recruitment and is therefore dominated by the control. An ICER is not presented here, as it offers limited additional value for decision-making in this context.

## Discussion

In this SWAT, we examined whether the use of a one-page infographic would improve recruitment and retention rates to the BASIL+ RCT. Our results show that a higher proportion of participants returned expressions of interest, were recruited and were retained in the no infographic group than in the infographic group, although the absolute differences were fairly modest and the odds ratios estimated from the adjusted analyses were not statistically significant.

These results are somewhat inconsistent with recent, positive evidence for the use of infographics in healthcare settings, particularly for improving return of expressions of interest [9]. The results suggest that although infographics may be a good way to convey information to potential participants, this may not necessarily translate to increased interest in the trial and subsequently increased recruitment or retention rates.

One potential explanation is that the older adult population approached for the BASIL+ trial was less responsive to infographics as a method for delivering information than participants of other ages included in other studies. However, this is not entirely convincing, as studies comparing the recall of information presented with and without graphics consistently shows that graphics increase recall for older adults as well as younger people [11].

**Table 3. Summary of SWAT economic analysis results.**

| Cost Type | Cost Description | Time Associated | Total Cost (£ in 2022)* |
|---|---|---|---|
| Design | Study team salary for designing the infographic | 660 minutes at average £22.173/hour | 243.90 |
| | PPI advisory group payment | 270 minutes at £29.7/hour | 133.88 |
| Printing and postage | Printing and posting of 6136 single-sided A4 portrait documents on 100gsm premium treated white paper were coordinated by a third-party service. | N/A | 443.76 (0.07 per document) |
| Total | | | 821.52 |

* The costs were adjusted to 2022 GBP values using Consumer Price Inflation (CPI) 2024 table from the Office for National Statistics (ONS). The inflation factors applied were £1 in 2020 = £1.19 in 2022, and £1 in 2021 = £1.11 in 2022.

Another suggestion is that the infographic included in our SWAT was not compelling enough to increase interest in the trial. For example, it may be that the SWAT infographic used in BASIL+ contained too much written text, reducing any positive impact of the visual representation of the study information. Previous SWAT research has suggested that using a professional graphic design service can lead to an increase in expressions of interest [9] and response rates [24]. The infographic used in the BASIL+ SWAT was designed and produced by trial team members in collaboration with PPI AG members with no professional expertise in graphic design. A professional graphic design service may have been able to make useful suggestions on the level of detail and visual graphics to be included in the infographic. A professionally produced infographic may therefore have a more meaningful impact on recruitment and retention rates for RCTs. This is an important consideration for future researchers, as using professional services carries a cost implication, although this cost may be offset by reduced input from internal team members and their associated salary costs.

The format of the main BASIL+ PIS may have been a factor in reducing the overall impact of the infographic in this SWAT. The PIS was printed in the form of 8 loose A4 sheets rather than in a booklet or stapled format. This may have caused the additional infographic to have become 'lost' within the other study documents rather than standing out as a separate information document. Indeed, it is plausible that the inclusion of an additional loose sheet of paper may have led some participants to have felt overwhelmed by all the study information, thus contributing to the lower number of expressions of interest from SWAT participants in the infographic group. The trial team were not able to control the order of the documents within the study information pack, which may have differed across SWAT participants and further increased the chances of the infographic being overlooked. Future researchers undertaking similar SWAT research may benefit from considering such production issues, especially where third-party involvement may be required.

Another consideration is the printed nature of the BASIL+ PIS. As outlined in Table 3, printing and postage costs constituted a large proportion of the total cost incurred for the SWAT intervention. Choosing an electronic PIS would likely eliminate much of this cost, potentially improving the cost effectiveness of our infographic, however the suitability of digital versus paper material for the target population remains an important factor. This is an important consideration for future researchers looking to replicate this SWAT.

The results of this SWAT have several implications for trialists undertaking RCTs in the future. For example, these results would suggest that adding an infographic alongside the main PIS may not be an effective use of resources when considering how to improve recruitment and retention to trials within a similar setting and population to BASIL+. However, there remains a scarcity of evidence on the use of infographics for improving recruitment and retention in RCTs, and further evaluations of this type of intervention are needed. As individual SWATs lack statistical power, replication of this SWAT is required to conclusively determine whether infographics are an effective tool for increasing trial recruitment and retention.

The SWAT was impeded by certain limitations, some of which were imposed due to Covid-19 restrictions in place during recruitment. For example, this SWAT was included in only one host trial. It would be advantageous to run this SWAT over multiple host trials in order to increase the sample size and therefore the statistical power of the SWAT results. In addition, incorporating participant feedback into the design of this SWAT may have produced richer results. For example, a qualitative element may have revealed insights into whether the infographic was an important factor in the decision to join and/or remain in the trial.

## Conclusion

When considering methods to improve recruitment and retention of an older adult population to an RCT, our results suggest that the use of a one-page infographic alongside the text-based participant information sheet may not be effective. This is a useful finding which, if replicated in future SWAT research, informs RCT design by highlighting potentially ineffective trial methods to be avoided.

Undertaking SWATs is a valuable way to evaluate recruitment and retention methods, and further research is needed to confirm these results.

## Supporting information

**S1 File. BASIL+ Infographic SWAT Dataset.**
(CSV)

## Acknowledgments

We would like to thank: the participants for taking part in the trial, general practice and North East and North Cumbria Local Clinical Research Network staff for identifying and facilitating recruitment of participants, the independent Programme Steering Committee members for overseeing the study and our PPI AG members for their insightful and valuable contributions to this project.

## Author contributions

**Conceptualization:** Eloise Ryde, Lauren Burke, Samantha Brady, Leanne Shearsmith, Andrew Henry, Della Bailey, Rebecca Woodhouse, Dean McMillan, Simon Gilbody, David Ekers, Elizabeth Littlewood.

**Data curation:** Eloise Ryde, Lauren Burke, Leanne Shearsmith, Charlie Peck, Han-I Wang.

**Formal analysis:** Charlie Peck, Han-I Wang.

**Funding acquisition:** Dean McMillan, Simon Gilbody, David Ekers, Elizabeth Littlewood.

**Investigation:** Eloise Ryde, Lauren Burke, Samantha Brady, Leanne Shearsmith, Andrew Henry, Rebecca Woodhouse, David Ekers.

**Methodology:** Eloise Ryde, Lauren Burke, Samantha Brady, Leanne Shearsmith, Andrew Henry, Rebecca Woodhouse, David Ekers, Elizabeth Littlewood.

**Project administration:** Eloise Ryde, Lauren Burke, Samantha Brady, Leanne Shearsmith, Andrew Henry, Rebecca Woodhouse.

**Resources:** Eloise Ryde, Lauren Burke, Samantha Brady, Leanne Shearsmith, Andrew Henry, Dean McMillan, Simon Gilbody, David Ekers, Elizabeth Littlewood.

**Supervision:** Kalpita Baird, Caroline Fairhurst, Dean McMillan, Simon Gilbody, David Ekers, Elizabeth Littlewood.

**Validation:** Kalpita Baird, Charlie Peck, Andrew Henry, Caroline Fairhurst, Han-I Wang.

**Visualization:** Eloise Ryde, Lauren Burke, Samantha Brady, Leanne Shearsmith, Andrew Henry, Della Bailey, Rebecca Woodhouse, Elizabeth Littlewood.

**Writing – original draft:** Lucy Atha.

**Writing – review & editing:** Lucy Atha, Eloise Ryde, Lauren Burke, Samantha Brady, Kalpita Baird, Leanne Shearsmith, Charlie Peck, Andrew Henry, Caroline Fairhurst, Han-I Wang, Della Bailey, Rebecca Woodhouse, Dean McMillan, Simon Gilbody, David Ekers, Elizabeth Littlewood.

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
