## [Decision Letter · Decision Letter 0]

16 Apr 2025

Investigating the use of a one-page infographic to improve recruitment and retention to the BASIL+ Randomised Controlled Trial: A Study Within a Trial (SWAT)

Dear Dr. Atha,

Thank you for submitting your manuscript to PLOS ONE. After careful consideration, we feel that it has merit but does not fully meet PLOS ONE’s publication criteria as it currently stands. Therefore, we invite you to submit a revised version of the manuscript that addresses the points raised during the review process.

We look forward to receiving your revised manuscript.

Kind regards,

Anna Bernasconi, PhD

Academic Editor

PLOS ONE

Additional Editor Comments (if provided):

Dear authors, your work was in general well received and appreciated by the reviewers, who recommend minor revisions. Please consider them carefully and return a revised version of the manuscript with clearly marked additions/changes and a response letter addressing each comment one by one. We look forward receiving this material for proceeding with the next assessment.

Reviewers' comments:

Reviewer's Responses to Questions

**Comments to the Author**

1. Is the manuscript technically sound, and do the data support the conclusions?

Reviewer #1: Yes

Reviewer #2: Yes

2. Has the statistical analysis been performed appropriately and rigorously?

Reviewer #1: Yes

Reviewer #2: Yes

3. Have the authors made all data underlying the findings in their manuscript fully available?

Reviewer #1: Yes

Reviewer #2: Yes

4. Is the manuscript presented in an intelligible fashion and written in standard English?

Reviewer #1: Yes

Reviewer #2: Yes

Reviewer #1: PONE-D-25-09949

Investigating the use of a one-page infographic to improve recruitment and retention to

the BASIL+ Randomised Controlled Trial: A Study Within a Trial (SWAT)

Reviewer’s report

Dear authors,

Your study investigates a relevant topic, is clearly structured, and is well written. I have just a few minor suggestions to enhance the quality of your manuscript.

Lines 141-142, bottom of page 8 and top of page 9: As PPI AG stands for Patient and Public Involvement Advisory Group the term ‘group’ is redundant and should be removed from line 142.

Results, lines 215-216: Could you explain why there is a different number of information packs sent out with and without the infographic (6,136 with the infographic and 5,808 without)?

Currently, Table 3 is located in the discussion section.

Reviewer #2: This is a well-reported and well-executed SWAT. It is also good to see an economic analysis embedded within a SWAT. I make the following observations/suggestions:

Introduction: Would it be possible to present the justification for the economic analysis in the background?

Outcomes: Describing outcomes as 'cost information' is somewhat vague - could you be more specific here, isn't it the ICER and the cost of the SWAT intervention (overall and per participant)? This might just require some relocation of material from the 'Economic Analysis' section.

SWAT Interventions: It is good also to see PPI included within the SWAT, and documentation of how this has led to the development of the infographic. Many readers would welcome even more specifics of the 'key information' and 'detail' that changed as a result of this. Given that you state PPI were 'heavily involved' please consider using the GRIPP2 short form or at least elaborating on some of the methods of involvement (also noting such barriers that may be relevant given that this research was conducted during Covid-19.) If PPI and GRIPP2 have been more extensively reported in the main trial, it may be appropriate to signpost to these as well. Doing the above will demonstrate that you have done meaningful PPI better than a statement to that effect.

Randomisation: I found the description of digital recording of allocation and attempts to achieve allocation concealment somewhat vague. Please clarify the method of digital record and who the team members/researchers were? Suggest revisiting lines 181-186 in particular especially with respect to clarifying what 'limited involvement' means.

Economic analysis: lines 247-8 possibly consider more cautious wording around effectiveness, given that SWATs are not powered to show effectiveness (though I fully accept that the infographic has a high certainty of being more costly).

Economic results. Table 3. To aid transparency, please consider reporting the resources used in terms of staff and PPI AG time rather than just the total cost?

Discussion: Consider acknowledging moves to use e-PIS and the impact that an e-infographic would have on costs given that a large proportion of the cost of the infographic was printing.

Optional note: The paragraphs beginning 301 and 306 may be interpreted as providing somewhat contradictory advice to triallists. I don't disagree with either statement, but it might be stronger to combine the paragraphs (the two thoughts are neatly contained in the conclusion).

**Do you want your identity to be public for this peer review?** For information about this choice, including consent withdrawal, please see our Privacy Policy

Reviewer #1: **Yes: ** Barbara Seebacher, PD, PhD

Reviewer #2: **Yes: ** Rebecca Kandiyali

---

## [Author Response · Author response to Decision Letter 1]

27 Jun 2025

Reviewer 1 Comments

• Comment 1: Lines 141-142, bottom of page 8 and top of page 9: As PPI AG stands for Patient and Public Involvement Advisory Group the term ‘group’ is redundant and should be removed from line 142.

Response: Thank you for this constructive feedback, the term ‘group’ has been removed from our manuscript.

• Comment 2: Results, lines 215-216: Could you explain why there is a different number of information packs sent out with and without the infographic (6,136 with the infographic and 5,808 without)?

Response: Thank you for this comment. The difference in the number of study information packs sent out is due to the cluster RCT design - research sites across the two SWAT groups did not necessarily recruit equal numbers of GP practices, and each GP practice is unlikely to have identified the same number of eligible SWAT participants. This means it would be highly unlikely that the same number of study information packs would have been sent out to eligible participants across the two SWAT groups. In order to clarify this, we have added text to the ‘sample size’ section which now reads as follows (lines 187 - 190):

“…The sample size for the SWAT depended on the number of participating research sites open to recruitment, the number of general practices engaging in participant recruitment activities via each research site, and the number of eligible SWAT participants who were sent the study information via their general practice.”

• Comment 3: Currently, Table 3 is located in the discussion section.

Response: Thank you for bringing this to our attention, we have moved Table 3 to the Results section (p15).

Reviewer 2 Comments

• Comment 1: Introduction: Would it be possible to present the justification for the economic analysis in the background?

Response: We thank the reviewer for this helpful suggestion. A justification for including the economic analysis has been added to the background section which now reads as follows (lines 97 – 104):

"... Despite the growing interest in the use of infographics in trials, no studies to date have thoroughly evaluated their cost-effectiveness. Given the potential resource implications of developing and implementing infographics at scale, it is important to assess whether any improvements in participant recruitment or retention justify the additional costs. In this SWAT, we therefore aimed to evaluate the effectiveness and cost-effectiveness of using a one-page infographic alongside the standard text-based PIS for improving recruitment and retention to the BASIL+ RCT (13, 14). ..."

• Comment 2: Outcomes: Describing outcomes as 'cost information' is somewhat vague - could you be more specific here, isn't it the ICER and the cost of the SWAT intervention (overall and per participant)? This might just require some relocation of material from the 'Economic Analysis' section.

Response: We thank the reviewer for this comment. We agree the term ‘cost information’ could be improved to provide more detail and have updated the manuscript which now reads as follows (lines 183 – 185):

“… SWAT intervention cost effectiveness was measured by the incremental cost per additional participant recruited, as well as the extra cost of the SWAT intervention overall and per-participant.”

• Comment 3: SWAT Interventions: It is good also to see PPI included within the SWAT, and documentation of how this has led to the development of the infographic. Many readers would welcome even more specifics of the 'key information' and 'detail' that changed as a result of this. Given that you state PPI were 'heavily involved' please consider using the GRIPP2 short form or at least elaborating on some of the methods of involvement (also noting such barriers that may be relevant given that this research was conducted during Covid-19.) If PPI and GRIPP2 have been more extensively reported in the main trial, it may be appropriate to signpost to these as well. Doing the above will demonstrate that you have done meaningful PPI better than a statement to that effect.

Response: We thank the reviewer for this helpful comment. We feel that the GRIPP2 short form was not necessary to use for our SWAT paper, however we have updated the manuscript to signpost to our main BASIL+ trial paper (Gilbody et al., 2024) which does provide more detailed information on the PPI involvement. We also plan to publish a manuscript specifically detailing how our PPI AG were involved in the BASIL+ trial, where the GRIPP will be used. We have updated the manuscript which now reads as follows (lines147 – 158):

“…The PPI AG was closely involved in the development of the infographic from concept to completion. Due to the social distancing restrictions in place at the time of designing the infographic, PPI AG meetings were conducted via video call. Initial meetings involved discussing the infographic idea and determining it’s format and content. Subsequent meetings focussed on refining drafts of the infographic, discussing the level of detail needed, as well as providing feedback on the colours and graphics used in the final document. For example, the addition of the arrows on the infographic was suggested by the PPI AG, as well as reducing the amount of text included. PPI AG members were reimbursed for their time according to INVOLVE guidelines (23). Details of PPI AG involvement in BASIL+ are provided in the main BASIL+ findings paper (14).”

• Comment 4: Randomisation: I found the description of digital recording of allocation and attempts to achieve allocation concealment somewhat vague. Please clarify the method of digital record and who the team members/researchers were? Suggest revisiting lines 181-186 in particular especially with respect to clarifying what 'limited involvement' means.

Response: We thank the reviewer for this helpful comment. We have provided further information on the digital recording method and provided further clarification on the roles of the team members. We have updated the manuscript which now reads as follows (lines 193 – 201):

“…BASIL+ research sites were randomised to a SWAT group on a 1:1 basis using simple randomisation in Stata v16 (StataCorp. 2019. Stata Statistical Software: Release 16. College Station, TX: StataCorp LLC.) before recruitment began. The BASIL+ trial statistician randomised the research sites (using pseudonymised trial site ID number rather than site name) in one batch in December 2020 and passed the allocations to the trial team to distribute study information packs according to SWAT allocation group. Knowledge of SWAT group allocation was limited to key members of the BASIL+ trial team who were not directly involved in BASIL+ participant recruitment or completion of trial follow ups.”

• Comment 4: Economic analysis: lines 247-8 possibly consider more cautious wording around effectiveness, given that SWATs are not powered to show effectiveness (though I fully accept that the infographic has a high certainty of being more costly).

Response: We thank the reviewer for this comment. We agree that more cautious wording around effectiveness is needed given the limitations of SWATs and have revised the text accordingly (lines 262 – 267):

"... The ICER of the SWAT intervention was calculated as −£136,920 per additional participant recruited (£821.52/ (3.6%-4.2%)), suggesting that, within the limitations of this SWAT, the infographic intervention may not be a preferred option, as it is likely to be more costly to implement without clear evidence of improved participant recruitment compared to the control."

• Comment 5: Economic results. Table 3. To aid transparency, please consider reporting the resources used in terms of staff and PPI AG time rather than just the total cost?

Response: We thank the reviewer for this suggestion and have updated Table 3 to show the ‘Time Allocated’ to the activity alongside the associated cost (p15).

• Comment 6: Consider acknowledging moves to use e-PIS and the impact that an e-infographic would have on costs given that a large proportion of the cost of the infographic was printing.

Response: We acknowledge and thank the reviewer for this important point and have updated the manuscript as follows (lines 322 – 328):

“…Another consideration is the printed nature of the BASIL+ PIS. As outlined in Table 3, printing and postage costs constituted a large proportion of the total cost incurred for the SWAT intervention. Choosing an electronic PIS would likely eliminate much of this cost, potentially improving the cost effectiveness of our infographic, however the suitability of digital versus paper material for the target population remains an important factor. This is an important consideration for future researchers looking to replicate this SWAT.”

• Comment 7: Optional note: The paragraphs beginning 301 and 306 may be interpreted as providing somewhat contradictory advice to triallists. I don't disagree with either statement, but it might be stronger to combine the paragraphs (the two thoughts are neatly contained in the conclusion).

Response: We are grateful for this valuable suggestion and have combined these paragraphs as suggested (lines 329 – 337).

---

## [Decision Letter · Decision Letter 1]

22 Jul 2025

Dear Dr. Atha,

Thank you for submitting your manuscript to PLOS ONE. After careful consideration, we feel that it has merit but does not fully meet PLOS ONE’s publication criteria as it currently stands. Therefore, we invite you to submit a revised version of the manuscript that addresses the points raised during the review process.

We look forward to receiving your revised manuscript.

Kind regards,

Anna Bernasconi, PhD

Academic Editor

PLOS ONE

Journal Requirements:

Additional Editor Comments:

Dear authors, thank you for providing the revised version of your manuscript. The two reviewers and myself have re-assessed it and found that the quality has been consistently improved. Please fix according to the minor comment of Rev#2

Reviewers' comments:

Reviewer's Responses to Questions

**Comments to the Author**

Reviewer #1: All comments have been addressed

Reviewer #2: All comments have been addressed

2. Is the manuscript technically sound, and do the data support the conclusions?

Reviewer #1: Yes

Reviewer #2: Yes

3. Has the statistical analysis been performed appropriately and rigorously?

Reviewer #1: Yes

Reviewer #2: Yes

4. Have the authors made all data underlying the findings in their manuscript fully available?

Reviewer #1: Yes

Reviewer #2: Yes

5. Is the manuscript presented in an intelligible fashion and written in standard English?

Reviewer #1: Yes

Reviewer #2: Yes

Reviewer #1: PONE-D-25-09949-R1

Investigating the use of a one-page infographic to improve recruitment and retention to

the BASIL+ Randomised Controlled Trial: A Study Within a Trial (SWAT)

Reviewer’s report

Dear Authors,

I appreciate the revisions made to your manuscript and recommend it for publication.

Kind regards

Reviewer #2: It is not normal practice to report an ICER when the intervention is dominated - suggest that you say something along the lines of...."The intervention was associated with higher costs (£821.52 per site) and a lower recruitment rate (3.6% vs 4.2%) than the control.

This indicates that the intervention was more costly and less effective, and therefore dominated by the control.

As a result, an ICER is not presented, as it would not provide meaningful decision-making information."

**Do you want your identity to be public for this peer review?** For information about this choice, including consent withdrawal, please see our Privacy Policy

Reviewer #1: No

Reviewer #2: **Yes: ** Rebecca Kandiyali

---

## [Author Response · Author response to Decision Letter 2]

5 Aug 2025

Reviewer 2 Comment

Comment: It is not normal practice to report an ICER when the intervention is dominated - suggest that you say something along the lines of...."The intervention was associated with higher costs (£821.52 per site) and a lower recruitment rate (3.6% vs 4.2%) than the control.

This indicates that the intervention was more costly and less effective, and therefore dominated by the control.

As a result, an ICER is not presented, as it would not provide meaningful decision-making information."

Response: We are grateful to Reviewer 2 for their constructive feedback. We agree that the inclusion of an ICER when the intervention is dominated adds little to the decision-making process. Despite this being part of our original outcome plan, we have removed the ICER as suggested and have reworded the text accordingly.

---

## [Editor Report · Decision Letter 2]

8 Aug 2025

Investigating the use of a one-page infographic to improve recruitment and retention to the BASIL+ randomised controlled trial: A Study Within a Trial (SWAT)

PONE-D-25-09949R2

Dear Dr. Atha,

We’re pleased to inform you that your manuscript has been judged scientifically suitable for publication and will be formally accepted for publication once it meets all outstanding technical requirements.

Kind regards,

Anna Bernasconi, PhD

Academic Editor

PLOS ONE

Additional Editor Comments (optional):

The manuscript can now be accepted in its current form after addressing all the comments of both reviwers and academic editor.
---

## [Editor Report · Acceptance letter]

PONE-D-25-09949R2

PLOS ONE

Dear Dr. Atha,

I'm pleased to inform you that your manuscript has been deemed suitable for publication in PLOS ONE. Congratulations! Your manuscript is now being handed over to our production team.

Kind regards,

on behalf of

Dr. Anna Bernasconi

Academic Editor

PLOS ONE